# Coevolution of religious and political authority in Austronesian societies

**Oliver Sheehan** [1] ✉, **Joseph Watts**[1,2,3], **Russell D. Gray**[1,4], **Joseph Bulbulia**[5,6], **Scott Claessens**[4], **Erik J. Ringen**[7] & **Quentin D. Atkinson** [1,4]

Authority, an institutionalized form of social power, is one of the defining features of the large-scale societies that evolved during the Holocene. Religious and political authority have deep histories in human societies and are clearly interdependent, but the nature of their relationship and its evolution over time is contested. We purpose-built an ethnographic dataset of 97 Austronesian societies and used phylogenetic methods to address two long-standing questions about the evolution of religious and political authority: first, how these two institutions have coevolved, and second, whether religious and political authority have tended to become more or less differentiated. We found evidence for mutual interdependence between religious and political authority but no evidence for or against a long-term pattern of differentiation or unification in systems of religious and political authority. Our results provide insight into how political and religious authority have worked synergistically over millennia during the evolution of large-scale societies.

Authority, a form of social power vested in a culturally recognized role or office and exercised over a specific group of people[1], is one of the defining characteristics of complex, large-scale societies. In small groups where a large proportion of members can interact directly, group decisions can be made on an informal and non-authoritarian basis. However, groups of more than a few thousand people generally require systems of command and control to make and implement group decisions[2,3]. During the Holocene, the scale and complexity of human societies increased immensely, and systems of authority became correspondingly more complex and ubiquitous[2,4].

A few small-scale societies reportedly lack authority altogether[5]. However, most societies (including those otherwise considered egalitarian) recognize authority at some level, minimally that of a household head over other household members[4,5]. In hierarchical societies, authority may be exercised over a sublocal group such as a clan or village ward, a local community such as a village or district, or a supralocal grouping such as a chiefdom or state, with higher levels of authority usually subsuming rather than replacing lower levels[2,3]. In addition to varying in its scope, authority varies in the domains of social life to which it applies. Many ethnographers distinguish between political (also 'secular', 'temporal' or 'civil') authority and religious ('ritual', 'sacred', 'spiritual' and so on) authority[6–8]. For some ethnographers, this distinction turns on the means by which authority operates, with political authority based on physical force and religious authority relying on supernatural sanctions or supernatural legitimacy[7,8]. Others make this distinction in terms of the ends to which authority is directed. Firth[6] describes politics as "focused on relations of men with other men", in contrast to religion, which "is more oriented to relations of men with gods or other spiritually conceived forces", and Garland[9] defines religious authority as "the right… to act authoritatively both in the name of and in matters of, religion". Here we use the ends-based distinction. We operationalize religious authority as a right to manage interactions between living human beings and supernatural agents or powers and political authority as a right to manage interactions between living human beings.

[1]Department of Linguistic and Cultural Evolution, Max Planck Institute for Evolutionary Anthropology, Leipzig, Germany. [2]Religion Programme, University of Otago, Dunedin, New Zealand. [3]Centre for Research on Evolution, Belief, and Behaviour, University of Otago, Dunedin, New Zealand. [4]School of Psychology, University of Auckland, Auckland, New Zealand. [5]School of Humanities, University of Auckland, Auckland, New Zealand. [6]School of Psychology, Victoria University of Wellington, Wellington, New Zealand. [7]Department of Anthropology, Emory University, Atlanta, GA, USA. ✉e-mail: oliver_sheehan@eva.mpg.de

Scholars acknowledge the historical interdependence of religion and politics, and by extension religious and political authority[6], but often emphasize one over the other. In many theories of political evolution, religion is downplayed, being either ignored altogether[10] or seen as merely underpinning or legitimizing existing political arrangements[1,11]. In others, religion is seen as foundational to politics, and religious authority is seen as the earliest form of authority[12–14]. Still others acknowledge a reciprocal relationship between religion and politics without assigning precedence to either. Religious expertise may be seen as one of multiple paths to power[15], or prosocial religious beliefs may be seen as having predisposed certain groups to evolve into large, complex societies[16]. A variant of this position is that religion and politics are so closely interwoven in most pre-modern societies that they cannot be meaningfully separated[17].

When religious and political authority are found in the same society, they can be differentiated to a greater or lesser degree. In many societies, they are combined in the same office (as in a polity headed by a priest-king or priest-chief), but in others religious and political power are wielded by distinct leaders who may cooperate or compete[6,18]. Many scholars have argued that the earliest forms of religious and political authority were combined, making distinct religious and political hierarchies a later development[14,19]. The opposing view that the earliest forms of religious and political authority were distinct is also encountered occasionally. The divine kingship of Hawaii, for example, has been explained as the outcome of a process whereby political leaders gained progressively more religious authority[20], presumably at the expense of more specialized religious figures. How religious and political authority have coevolved and whether there are historical regularities in their pattern of differentiation and fusion are separate but related questions, since one of the most obvious ways for religious and political authority to coevolve would be for both forms of authority to be vested in the same office.

Archaeological and historical evidence suggests answers to both questions. Archaeologists note that in most early city-states, the earliest monumental structures appear to have served religious rather than secular purposes[14,21,22], suggesting that religious authority may have come first. The extent to which the earliest forms of authority were differentiated is more difficult to infer from the archaeological record. However, the earliest written records clearly indicate that in at least one area of primary state formation, Mesopotamia, religious and political authority were initially combined and later became partly distinct[21]. Presently, the incompleteness of the historical and archaeological records leaves these questions unresolved.

Cultural phylogenetic methods can complement the archaeological record by using ethnographic data to infer the evolutionary histories of cultural traits—a technique called "virtual archaeology"[23]. These methods typically rely on language trees or phylogenies to model cultural ancestry[24]. Since there is no widely recognized phylogeny of the world's languages, cultural phylogenetic studies usually focus on cultural variation within a single recognized language family. The Austronesian language family of Southeast Asia and the Pacific has proved particularly well suited to a cultural phylogenetic approach. It is the second-largest language family whose taxonomic status is uncontroversial[25], and the societies that speak Austronesian languages are remarkably diverse. As well as having a great variety of social and political structures[26], the Austronesian-speaking world was until recently home to a large number of indigenous religions that were similarly diverse and are relatively well documented[27]. Because of these advantages, there have already been a number of cultural phylogenetic studies of Austronesian-speaking societies[28–30], including at least two that have examined the coevolution of socio-political phenomena and elements of religious belief and practice[31,32].

Here we present a cultural phylogenetic study of the evolution of religious and political authority in the Austronesian-speaking world. On the basis of ethnographic descriptions, we coded 97

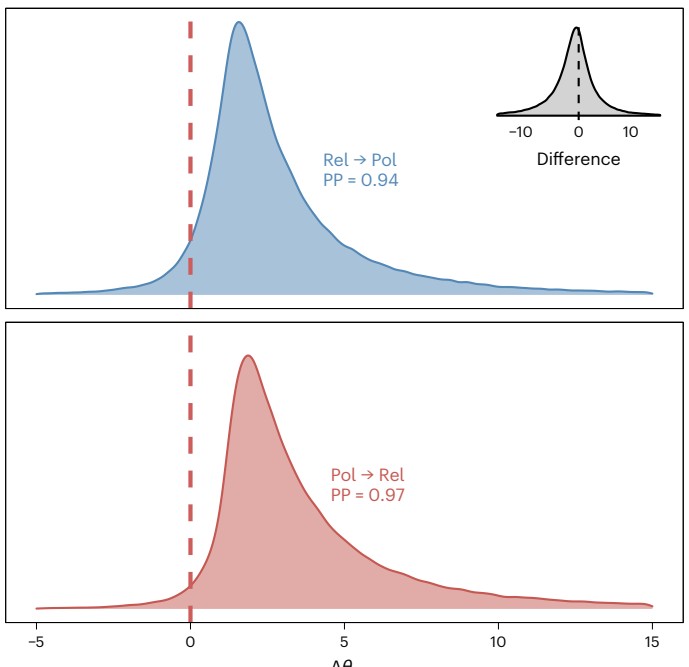

**Fig. 1 | Posterior distribution showing probability densities of changes in the equilibrium trait value $\theta$ of political authority (Pol) and religious authority (Rel) in response to a standardized unit increase in the other trait.** Posterior probabilities (PPs) denote the positive posterior mass (that is, the probability), given the model and the data, that an increase in political authority leads to an increase in religious authority, and vice versa. The values were scaled by the median absolute deviation, which is less sensitive to outliers than the standard deviation. The grey inset represents the posterior difference between the two distributions.

Austronesian-speaking societies with respect to whether they had systems of religious and/or political authority and, if applicable, the scale of the social groups that these systems encompassed. In societies in which both religious and political authority were present, we also coded the extent to which the two were differentiated. These variables were coded on four-point ordinal scales (Methods). We mapped the traits onto trees representing relationships between the languages spoken in these societies and reconstructed their evolutionary histories under different model assumptions to infer causal dependencies and patterns of differentiation.

## Results
### Coevolution
Our first series of phylogenetic analyses focused on the coevolution of religious and political authority. We coded both religious and political authority as ordinal variables with four possible states: absent (not present above the household level), sublocal (incorporating a group larger than the household but smaller than the local community), local (incorporating the local community) and supralocal (incorporating more than one local community). Both of these variables showed high phylogenetic signal ($\lambda$) (political authority: $\lambda = 0.58$; 95% highest posterior density interval (HPDI), (0.00, 0.80); religious authority: $\lambda = 0.55$; 95% HPDI, (0.00, 0.78); Extended Data Fig. 1) and were positively phylogenetically correlated (phylogenetic correlation, 0.78; 95% HPDI, (0.25, 0.99); residual correlation, 0.20; 95% HPDI, (−0.56, 0.94); Extended Data Fig. 2), suggesting that their evolution could reasonably be modelled as a dynamic coevolutionary process. Previous approaches to testing for the coevolution of cultural traits have only allowed the use of binary variables[30–32], resulting in a loss of information and hence statistical power. Here we overcome this limitation by assuming that each of the

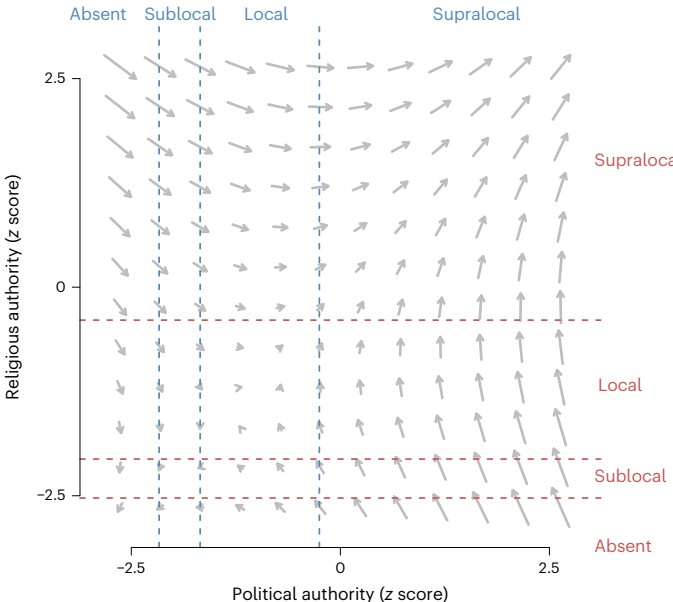

**Fig. 2 | Phase plane showing the expected change in levels of political and religious authority depending on the state of the other trait.** The authority variables are on the latent scale. The dashed lines are posterior median ordinal cutpoints, and the spaces between these represent the expected authority levels: absent, sublocal, local and supralocal. The grey arrows indicate the direction (orientation) and strength (size) of selection.

ordinal variables represents a latent continuous trait and modelling their coevolution using a recently developed Bayesian phylogenetic method that allows inferences to be made about the influence of two or more traits of any distribution on each other, as well as the role of 'selection' and 'drift' in the evolution of each[33]. Since linguistic distances between societies were positively correlated with geographic distances ($r$ = 0.31; 95% confidence interval, (0.28 0.33); d.f. = 4,654; $P < 0.001$; Extended Data Fig. 3), we adjusted for geographic distance in our model to mitigate any confounding effects of cultural diffusion or similar environments. Analysis of simulated data indicated that this model was able to accurately recover true parameter values (Extended Data Fig. 4), and standard post-analysis checks suggested that the model converged normally (Extended Data Fig. 5).

We found evidence for a reciprocal coevolutionary relationship between religious and political authority. Figure 1 presents the posterior change in the equilibrium trait value of one trait resulting from an absolute deviation increase in the other trait, and vice versa. Given the model, the data and our priors, we can be 97% certain that an absolute deviation increase in political authority results in an increase in religious authority at equilibrium (median posterior value, 2.44; 95% HPDI, (−0.03, 4.87); log Bayes factor (BF), 4.62). Similarly, we can be 94% certain that an absolute deviation increase in religious authority results in an increase in political authority at equilibrium (median posterior value, 2.00; 95% HPDI, (−0.53, 4.65); log BF, 2.84). We found no evidence of a difference between these two distributions (median posterior difference, −0.54; 95% HPDI, (−4.14, 3.63); log BF, −0.50) and hence no clear evidence that either form of authority had precedence.

Further inspection of the model dynamics revealed that combinations involving high levels of one trait and low levels of the other were unstable (Fig. 2 and Extended Data Fig. 6). When religious authority was low and political authority was high, there was strong positive selection on religious authority and negative selection on political authority. Similarly, when political authority was low and religious authority was high, there was strong positive selection on political authority and negative selection on religious authority. These model

dynamics entail runaway selection for each type of authority, such that authority levels enter a positive feedback loop and do not return to any stable equilibrium.

The coevolutionary model also illuminates the evolution of political and religious authority over time, providing estimated probabilities of different authority levels for ancestral nodes in the Austronesian language phylogeny (Fig. 3 and Extended Data Fig. 7). On the basis of our analysis, local political and religious authority is the most likely state for Proto-Austronesian society. In the more recent Proto-Central Pacific node, the probability of supralocal religious and political authority increases, and it becomes the most likely state in Proto-Polynesian. These reconstructions are consistent with previous work that has reconstructed the evolutionary history of political complexity in the Austronesian world[28] and the socio-religious system of Proto-Polynesian society specifically[34].

## Sequential evolution

In our second series of phylogenetic analyses, we tested for patterns of differentiation and fusion in systems of religious and political authority. We coded the structure of religious and political authority as one of four possible states: none (one or both forms of authority lacking above the household level), combined (vested in the same office or offices), partly independent (for example, vested in distinct offices that are part of the same hierarchy) and independent (vested in distinct offices that are not part of the same hierarchy). We tested four sequential models of trait evolution against a full model that allowed any transition between any level of differentiation (Fig. 4). Two of the sequential models required more differentiated authority structures to evolve from less differentiated ones. These differentiation models consisted of a strong version and a weak version. In the strong version, independent could evolve only from partly independent, and partly independent could evolve only from combined, whereas the weak version also allowed a direct transition from combined to independent. The other two sequential models required less differentiated authority structures to evolve from more differentiated ones. These unification models also consisted of a strong version (which required combined to evolve from partly independent, and partly independent to evolve from independent) and a weak version (which also allowed a direct transition from independent to combined). We evaluated the various models by comparing log BFs calculated from the log marginal likelihoods estimated by the analyses. The results were equivocal: no model outperformed any of the others (Supplementary Table 1).

## Discussion

We found evidence for a reciprocal coevolutionary dependency between religious and political authority in our sample. The relationship could have been direct or could have been the result of a third variable, which may have corresponded to a higher-level concept encompassing both religious and political authority. Regardless, we found no evidence that either form of authority had causal precedence. Furthermore, we did not find any evidence for long-term patterns of differentiation or fusion in systems of religious and political authority.

A direct coevolutionary relationship between religious and political authority seems highly plausible given ethnographic descriptions of the two institutions being closely intertwined. In many Austronesian societies, supreme religious and political authority were vested in the same office, the divine kingship of Hawaii being perhaps the best-known example[35]. In others there was a partial separation of religious and political authority. The details varied. The two institutions might be vested in different offices that were part of the same hierarchy. In Tonga, the priestly Tu'i Tonga outranked more powerful political rulers[36], whereas in Roviana, 'chiefs' (*bangara*) enjoyed supremacy over 'high priests' (*ngati hiama*) except in religious matters[37]. Other arrangements were less straightforward. Tikopia was ruled by four chiefs (*ariki*) who had equal political status, but one of

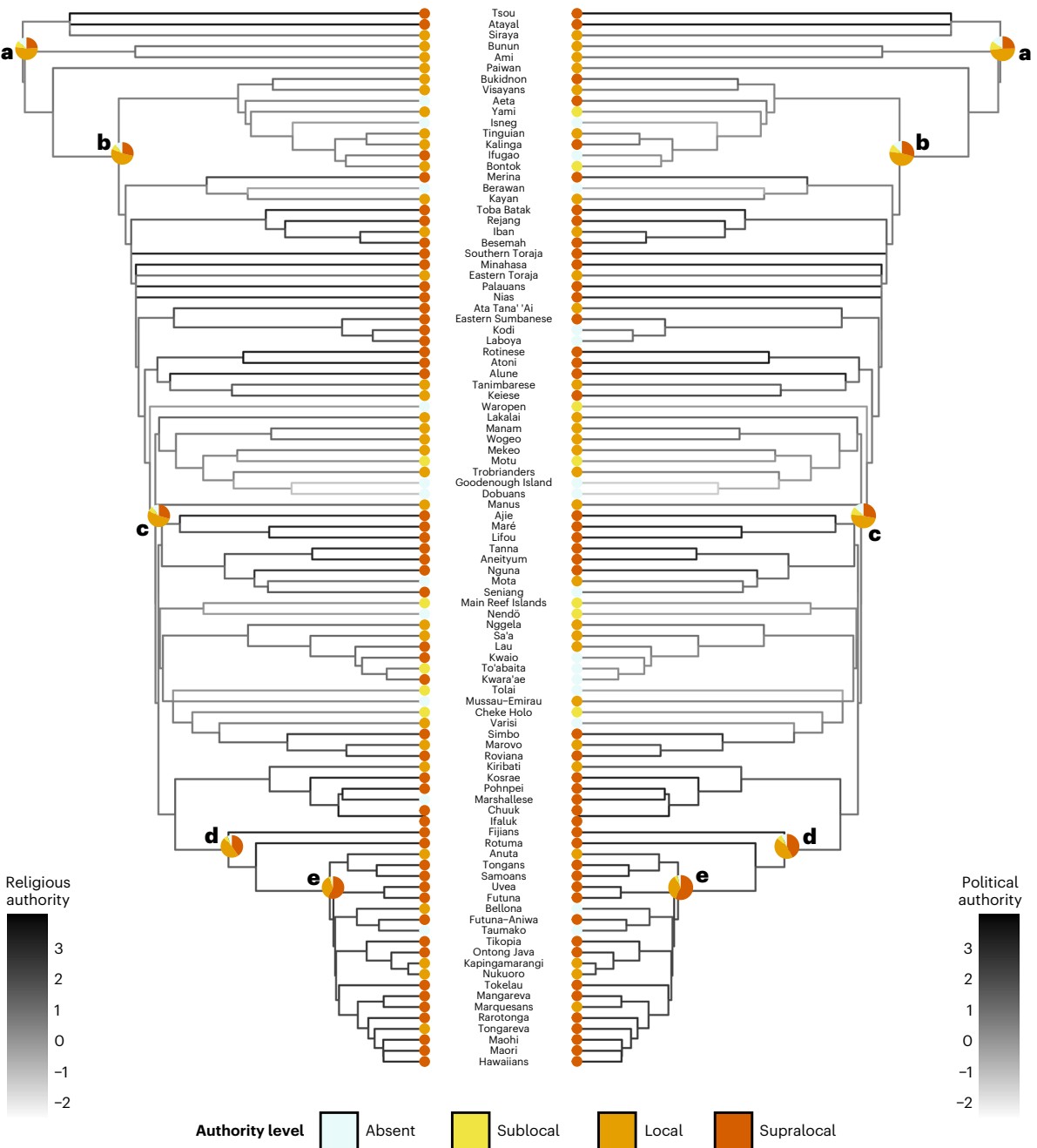

**Fig. 3 | Observed data and ancestral state reconstructions mapped onto a maximum clade credibility tree for the Austronesian language phylogeny.** The colours at the tips represent the observed levels of religious authority (left) and political authority (right) across 97 Austronesian societies. The shading of the branches represents the median posterior values of the continuous latent variables at each ancestral node, implied by the coevolutionary model. Darker shading indicates higher levels of authority in ancestral state reconstructions. The pie charts show the median posterior probabilities of each authority level for the following ancestral nodes: Proto-Austronesian (**a**), Proto-Malayo-Polynesian (**b**), Proto-Oceanic (**c**), Proto-Central-Pacific (**d**) and Proto-Polynesian (**e**). These five nodes were selected on the basis of previous work (see, for example, ref. [35]).

these chiefs (the Ariki Kafika) was "in island-wide religious ceremonies…clearly pre-eminent"[38]. Even in societies where religious and political leaders enjoyed de jure independence and were sometimes opposed, they usually headed the same social group and often worked together closely. In Tahiti, for example, high priests are reported to have "exercised immense influence" in secular affairs, "depending more or less on the character of the king"[39]. The ethnographic sources often explicitly describe religious authority as supporting political authority by legitimizing it and reinforcing it with supernatural sanctions. In Chuuk, for example, the *itang* ('political priests') "legitimized chiefship through divine sanction and the spirit power (*manaman*) that went with it", according to one source[40]. Given the centrality of religious belief and practice in pre-modern societies[13], it seems likely that aspiring political leaders who lacked either religious authority of their own or the support of religious leaders would have struggled to gain and maintain power. It is less obvious why religious authority would have depended so heavily on political authority, but political support might have strengthened religious authority by increasing its prestige and resource base and perhaps also by helping suppress challenges to its monopoly.

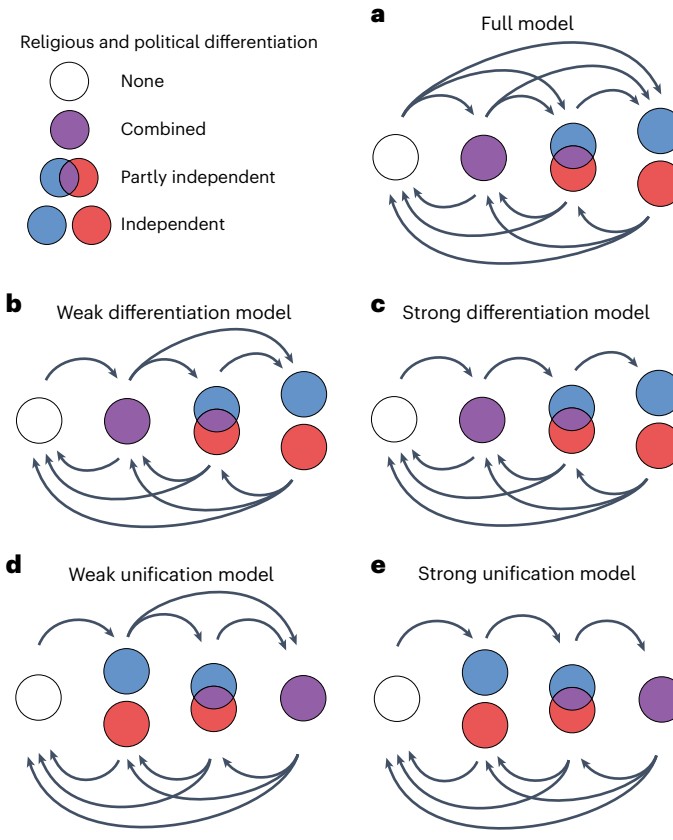

**Fig. 4 | Five models of the evolution of religious and political authority. a**, In the full model, any transition between any two states is allowed. **b**, In the weak differentiation model, independent systems of authority must evolve from either combined or partly independent systems, and partly independent systems must evolve from combined systems. **c**, In the strong differentiation model, independent systems of authority must evolve from partly independent systems of authority, which in turn must evolve from combined systems of authority. **d**, In the weak unification model, combined systems of authority must evolve from either independent or partly independent systems of authority, and partly independent systems of authority must evolve from independent systems of authority. **e**, In the strong unification model, combined systems of authority must evolve from partly independent systems of authority, which in turn must evolve from independent systems of authority.

The interdependence that we observed could also reflect a third variable that simultaneously caused changes in both political and religious authority. Authority itself—that is, a higher-level phenomenon encompassing both religious and political authority—is perhaps the most likely candidate. There are obvious reasons why a society with any given form of authority might have been more likely to gain (and less likely to lose) any other form. New forms of authority could have been vested in existing offices rather than requiring the creation of new ones, and existing forms of authority could have been transferred from defunct offices to remaining offices, increasing the redundancy in the system and reducing the chance of specific forms of authority being lost. Existing forms of authority could also have served as models for new ones, and populations that had already accepted one form of authority might well have been more willing to accept others. Social (or cultural) complexity, an even more encompassing phenomenon that may or may not represent a single underlying construct[41–43], is another plausible third variable. Environmental variables such as circumscription or resource concentration[10] could have played a role, though the fact that we controlled for geographic distance makes this seem less likely.

The interdependence between religious and political authority observed in the present study and its apparent lack of directionality are in keeping with the results of two previous studies that examined the coevolution of religious beliefs and practices with other socio-political traits and found evidence of reciprocal relationships[31,32]. Nevertheless, while our results do not support a directional relationship, the limitations of our data prevent us from ruling it out entirely. The coding of religious and political authority as ordinal variables with four states is likely to have made religious and political authority correspond somewhat more closely than they did in actuality. For example, the Toba Batak were coded as having both supralocal religious and supralocal political authority. However, although both religious and political authority existed on a supralocal level among the Toba Batak, the scope of religious authority was much greater. Some powerful chiefs governed groups of villages with combined populations of up to a thousand, but tens or perhaps even hundreds of thousands acknowledged the religious authority of the priest-king Si Singamangaraja[44]. Had a finer-grained coding system been feasible, it is possible that more evidence for directionality would have been observed.

The lack of support for patterns of differentiation or fusion in systems of religious and political authority may partly reflect sample characteristics. The fact that all or almost all early states had combined systems of political and religious authority[22] suggests that if there is a sustained trend towards differentiation, this trend emerges only in societies that have already reached the state level. Pre-colonial Austronesian societies varied greatly in their complexity, but there were few state-level societies[45] and fewer still among those who retained their indigenous religions until the modern era.

The societies in our sample represent only a fraction of the total number of Austronesian societies, and some areas (for example, Vanuatu) are undersampled relative to others (for example, Polynesia). We cannot be certain that our results generalize to the Austronesian-speaking world as a whole, let alone to the rest of the world. Nevertheless, the diversity of the societies in the sample, which occupy all corners of the Austronesian-speaking world and range from acephalous to state-level, gives us some confidence that the interdependence we observe is a real and general phenomenon. Further research could, of course, test the extent to which our findings apply elsewhere.

The present study found evidence for a reciprocal coevolutionary relationship between religious and political authority in the Austronesian-speaking world. This relationship could have been direct, caused by a third factor or both. We found no clear evidence for or against a progression from less differentiated to more differentiated systems of authority. Our results suggest that theories of cultural evolution that ignore or sideline religion are incomplete. Although many authors have argued that religious and political authority have coevolved, the present study provides quantitative evidence of the closeness of this relationship as well as specific insights into how these two institutions have worked synergistically during the evolution of large-scale societies.

## Methods

### Phylogenies

We modelled cultural ancestry using a sample of 1,000 trees from the posterior distribution of a previously published Bayesian reconstruction of the Austronesian language family. This set of trees originally included 400 taxa, 363 of which corresponded to unique Austronesian languages[29]. Of these languages, 109 corresponded to one of the 97 societies in our ethnographic dataset. None corresponded to more than one society. Only eight societies (Atayal, Bontok, Ifugao, Minahasa, Nendö, Tanimbar, Tanna and Visayans) corresponded to more than one language. In these cases, we took the conservative approach of selecting only one language per society, choosing the language with the greatest number of speakers according to Ethnologue[46]. The pruning of phylogenies employed the packages ape[47] and geiger[48] in the

programming language R[49]. The pruned set of phylogenies is available via the Open Science Framework (https://osf.io/cm53v/).

## Coding of variables

We coded 97 Austronesian-speaking societies with respect to three variables: religious authority, political authority and the structure of religious and political authority. Authority was defined as a form of social power vested in a specific social role or office and exercised over a specific group of people[3]. Religious authority was defined as a right to manage interactions between living human beings and supernatural agents or powers, whereas political authority was defined as a right to manage interactions between living human beings[6,9]. The variables 'religious authority' and 'political authority' each had the same four states. Societies in which the relevant form of authority did not exist or encompassed a group no larger than the household were coded 0 (authority at the household level was ignored partly because of its near-universality and partly because of the difficulty inherent in separating de jure authority from de facto power at the household level). Societies in which the relevant form of authority existed above the household level were coded 1 if the group it incorporated was sublocal (smaller than the local community), 2 if the group was local (coextensive with the local community) or consisted of multiple sublocal groups, and 3 if the group was supralocal (consisting of more than one local community). The local community was defined as "the maximal group of persons who normally reside together in face-to-face association"[50].

The variable 'structure of religious and political authority' represented the extent to which religious and political authority were differentiated. Societies were coded 0 if religious or political authority or both were lacking above the household level. If supreme religious and political authority were combined (vested in the same office or offices), the society was coded 1. Societies in which supreme religious and political authority were partly independent were coded 2. This was something of a residual category that included societies in which the two forms of authority were incompletely partitioned between different offices (for example, supreme political authority being vested in one office and supreme religious authority shared between this office and another) as well as those in which they were vested in different offices that were part of the same hierarchy (for example, a high priest being the subject of a secular high chief, or vice versa). Finally, societies in which supreme religious and political authority were independent (vested in different offices that were not part of the same hierarchy) were coded 3.

Austronesian societies have undergone dramatic changes in their religious and political organization through contact with non-Austronesian societies, particularly over the past few centuries. Almost all Austronesian societies underwent some form of colonization that resulted in permanent changes to their political systems. Moreover, almost all Austronesian speakers now affiliate with either Christianity or Islam, which have either replaced or supplemented their traditional religious beliefs and practices[51]. The cultural phylogenetic methods used in the present study assume predominantly vertical (within-lineage) cultural transmission[52], and so applying them to ethnographic data from Austronesian societies today is unlikely to be informative and could well be misleading. Hence, societies were coded as they were immediately prior to colonization and/or large-scale conversion to a world religion (whichever occurred earlier). Coding was based on a range of ethnographic sources. The data, along with citations and detailed notes justifying each coding decision, are provided in the most recent version of Pulotu[27], a database of Austronesian religions.

## Assessing phylogenetic signal and correlation

We assessed the strength of phylogenetic signal for political and religious authority (that is, the proportion of variance captured by phylogeny[53]), as well as the phylogenetic correlation between these variables, using a Bayesian phylogenetic generalized linear mixed model (see the Supplementary Methods for the full model formula). For this model, we used generic, weakly regularizing priors to impose conservatism on parameter estimates and facilitate model convergence. We iterated the model over 100 randomly drawn posterior trees. The model was fitted in R v.4.0.2 (ref. [49]) with the brms package[54] running Stan[55]. Standard Markov chain Monte Carlo (MCMC) diagnostics ($\hat{R} \leq 1.05$) and trace plots suggested that the model converged normally.

## Dynamic coevolutionary model

While our phylogenetic generalized linear mixed model indicated a phylogenetic correlation between political and religious authority, this static model could not distinguish directionality or contingencies in coevolution. To give us more insight into how these two variables have coevolved, we used a dynamic model of cultural change over the phylogenetic tree. Many authors have implemented this approach using the Discrete component of the software package BayesTraits[56], but that method is limited to binary traits. To avoid having to dichotomize our ordinal variables, we used a recently developed Bayesian method for dynamic coevolutionary analyses that can accommodate any number of traits of any distribution[33]. With this approach, ordinal traits are modelled as latent continuous variables evolving under selection (both autoregressive selection and cross-trait selection) and drift, similar to a multivariate Ornstein–Uhlenbeck model. The estimation of continuous latent authority levels in the model does not necessarily assume sequential evolution from one authority state to the next, since more than one observed authority state can be consistent with the same latent authority level. The parameters representing selection are used to derive standardized, directed measures of the strength of coevolution between variables ($\Delta\theta_z$, as shown in Fig. 1).

As in our assessment of phylogenetic signal, we used generic, weakly regularizing priors. We iterated the model over 100 randomly drawn posterior trees. We additionally included a Gaussian process with longitude and latitude values for each society to control for spatial proximity. The model was fitted in R v.4.0.2b[48] with the rstan package running Stan[54]. Standard MCMC diagnostics ($\hat{R} \leq 1.01$) and trace plots suggested that the model converged normally (Extended Data Fig. 5). Log BFs were computed for individual parameters by doubling the natural logarithm of the BF, computed with the bayestestR package[57].

## Simulations of the dynamic coevolutionary model

We ran simulations to confirm that our coevolutionary model could capture the true parameter values. We fixed several parameters in the model (specifically, parameters reflecting the strength of selection and drift) and generated 100 simulated datasets. Next, we fitted the coevolutionary model to each of these datasets and determined whether the 95% credible intervals for the posterior distributions contained the true parameter values. The results of the simulations showed that the coevolutionary model adequately recovered true parameter values (Extended Data Fig. 4).

## Sequential evolution

Models of sequential evolution were tested using the Multistate component of the software package BayesTraits (v.3.0)[56]. Multistate tests model the evolution of a single trait that adopts two or more discrete states, and they can be run using either a maximum likelihood or MCMC approach. The analyses reported in the present study used an MCMC approach, but the choice of priors was guided by preliminary analyses involving a maximum likelihood approach.

**Maximum likelihood estimations.** One hundred optimization attempts were made for each tree in the sample.

**MCMCs.** Each MCMC involved 100,000,000 iterations of the chain, with the first 10,000,000 removed as burn-in. On the basis of the results of the maximum likelihood estimations, a reverse-jump hyperprior

with an exponential distribution that can range between 0 and 10 was chosen for all analyses. A stepping-stone sampler with 100 stones was run for 100,000 iterations to estimate the log marginal likelihoods for the models in the posterior distribution of each analysis. All analyses were independently replicated three times, and each replication converged on highly similar rate and log marginal likelihood values (Supplementary Table 1).

Five models were tested (Fig. 4). In the full model, any transition between any two states was allowed. This allowed the analyses to select from all possible model structures. In the strong differentiation model, rates of transition from 0 to 2 (q02), 0 to 3 (q03) and 1 to 3 (q13) were set to zero. This constrained the analyses to include only models in which more differentiated authority structures evolved from less differentiated ones (1 → 2 → 3). In the strong unification model, rates of transition from 0 to 1 (q01), 0 to 2 (q02) and 3 to 1 (q31) were set to zero. This constrained the analyses to include only models in which less differentiated authority structures evolved from more differentiated ones (3 → 2 → 1). Since 2 (partly independent) is a more heterogeneous category than 0, 1 and 3, less stringent (weak) versions of the differentiation and unification models were also tested. In the weak differentiation model, only rates q02 and q03 were restricted to zero (that is, transitions from 1 to 3 were also allowed). In the weak unification model, only q01 and q02 were restricted to zero (that is, transitions from 3 to 1 were also allowed).

**Model comparison.** Support for the posterior distribution of analyses with different model structures was evaluated using log BFs calculated from the log marginal likelihoods obtained for each posterior distribution of models. Log BFs were interpreted following a scheme in which 0–2 is 'not worth more than a bare mention', 2–6 is 'positive evidence', 6–10 is 'strong evidence' and 10 or higher is 'very strong evidence'[58].

### Reporting summary
Further information on research design is available in the Nature Research Reporting Summary linked to this article.

## Data availability
The data are publicly available on Pulotu[27] as well as the Open Science Framework (https://osf.io/cm53v/).

## Code availability
The code and command files for all phylogenetic analyses are provided on the Open Science Framework (https://osf.io/cm53v/).

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

## Acknowledgements

O.S. was funded by a University of Auckland Faculty Research and Development Fund grant and a Rutherford Discovery Fellowship to Q.D.A. (no. RDF-UOA-1101). S.C. was funded by a Marsden Fund grant to Q.D.A. (no. 20-UOA-123). J.W. receives funding from the Marsden Foundation of New Zealand (grant no. 19-UOO-1932). J.B. was funded by a Templeton Religion Trust Grant (no. 0196). The funders had no role in study design, data collection and analysis, decision to publish or preparation of the manuscript.

## Author contributions

J.W., O.S. and Q.D.A. designed the study with input from J.B. and R.D.G. J.W., O.S. and S.C. performed the statistical analyses with input from E.J.R. and Q.D.A. O.S. wrote the manuscript with input from all other authors.

## Funding

## Competing interests

The authors declare no competing interests.

## Additional information

**Extended data** is available for this paper at https://doi.org/10.1038/s41562-022-01471-y.

**Correspondence and requests for materials** should be addressed to Oliver Sheehan.

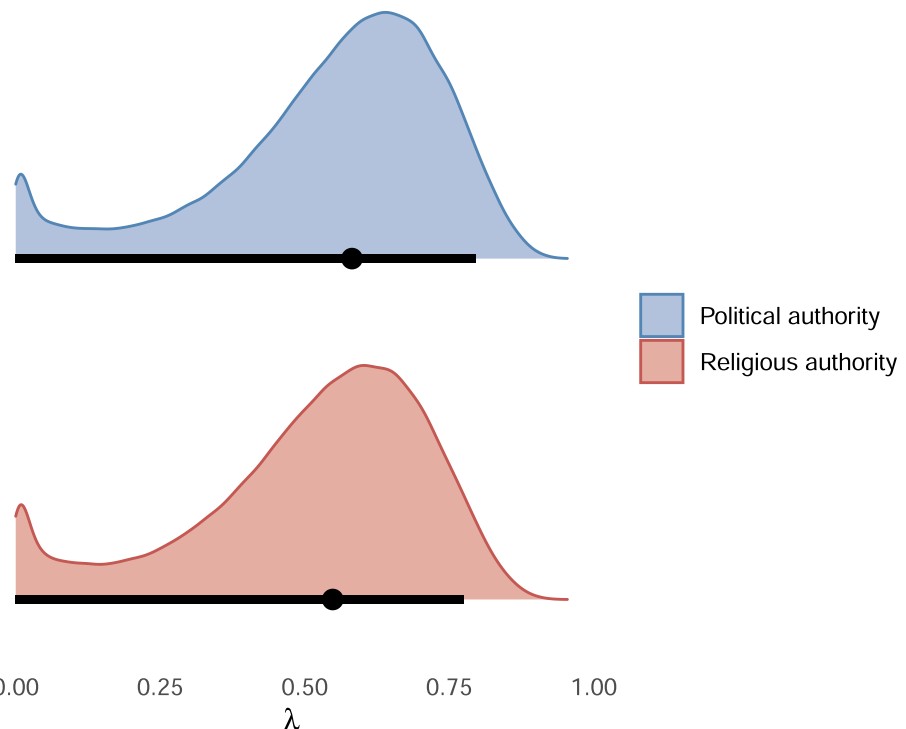

**Extended Data Fig. 1 | Phylogenetic signal (λ) for political and religious authority, as estimated by a Bayesian phylogenetic generalised linear mixed model.** Densities are posterior distributions for phylogenetic signal, points are posterior medians, and lines are 95% highest posterior density intervals (HPDI).

The phylogenetic signal for political authority was 0.58, 95% HPDI [0.00, 0.80], and the phylogenetic signal for religious authority was 0.55, 95% HPDI [0.00, 0.78], suggesting that these variables were suitable for coevolutionary analyses.

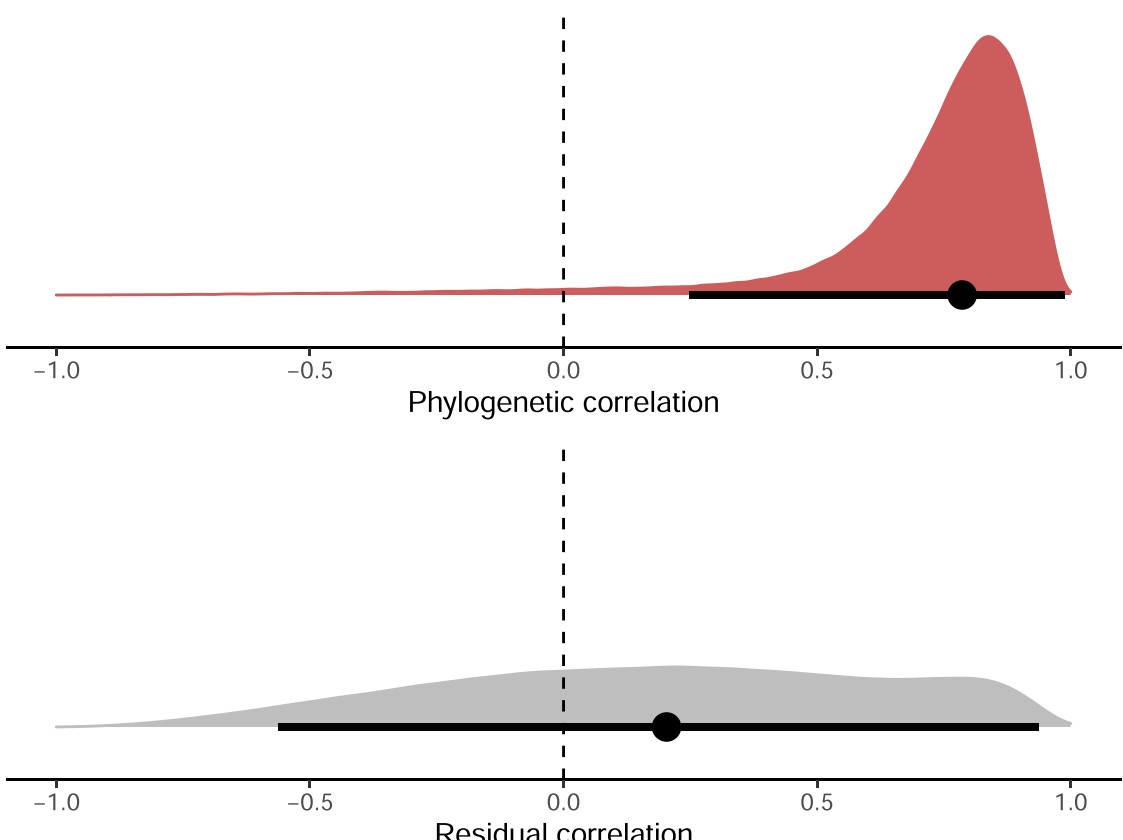

**Extended Data Fig. 2 | Phylogenetic and residual correlations between political and religious authority, estimated simultaneously in a Bayesian phylogenetic generalised linear mixed model.** Densities are posterior distributions for correlations, points are posterior medians, and lines are 95% highest posterior density intervals (HPDI). The phylogenetic correlation between the two types of authority was 0.78, 95% HPDI [0.25, 0.99], while the residual correlation was 0.20, 95% HPDI [−0.56, 0.94], suggesting that these variables were suitable for coevolutionary analyses.

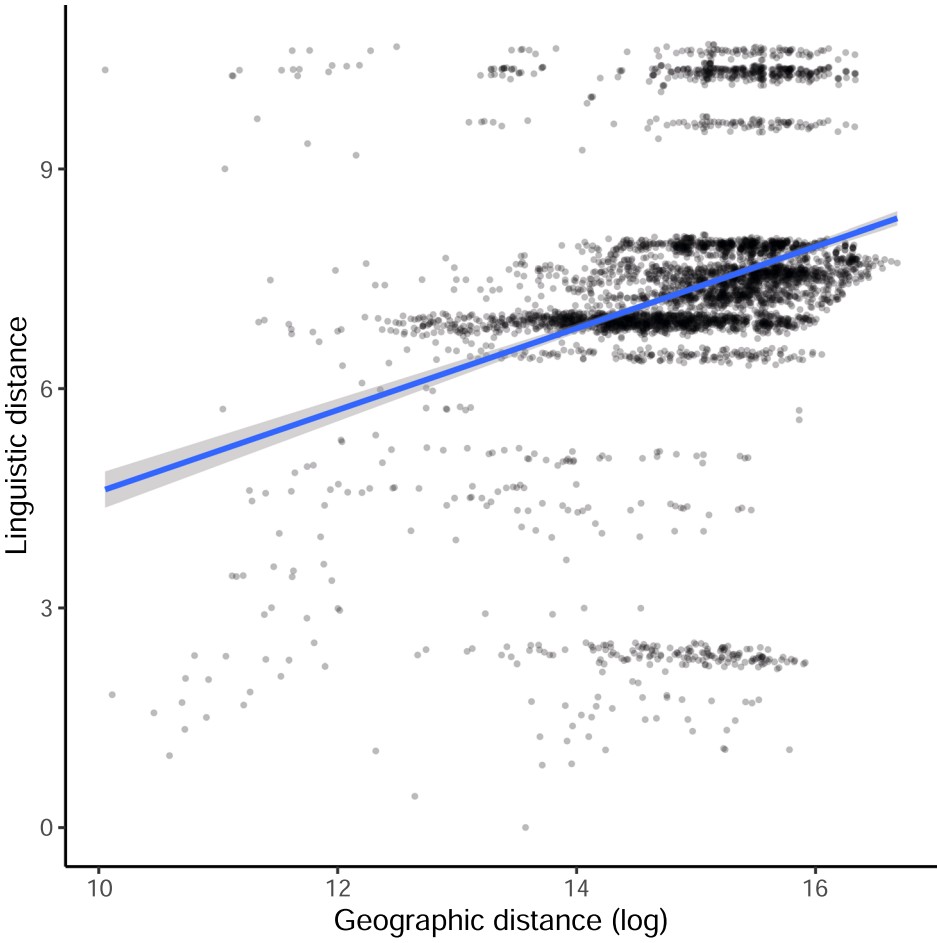

**Extended Data Fig. 3 | The relationship between linguistic and log geographic distances across 97 Austronesian societies.** Each point represents a pairwise relationship between two societies, resulting in 4,656 unique pairings. The line is a fitted linear regression with shaded 95% confidence intervals.

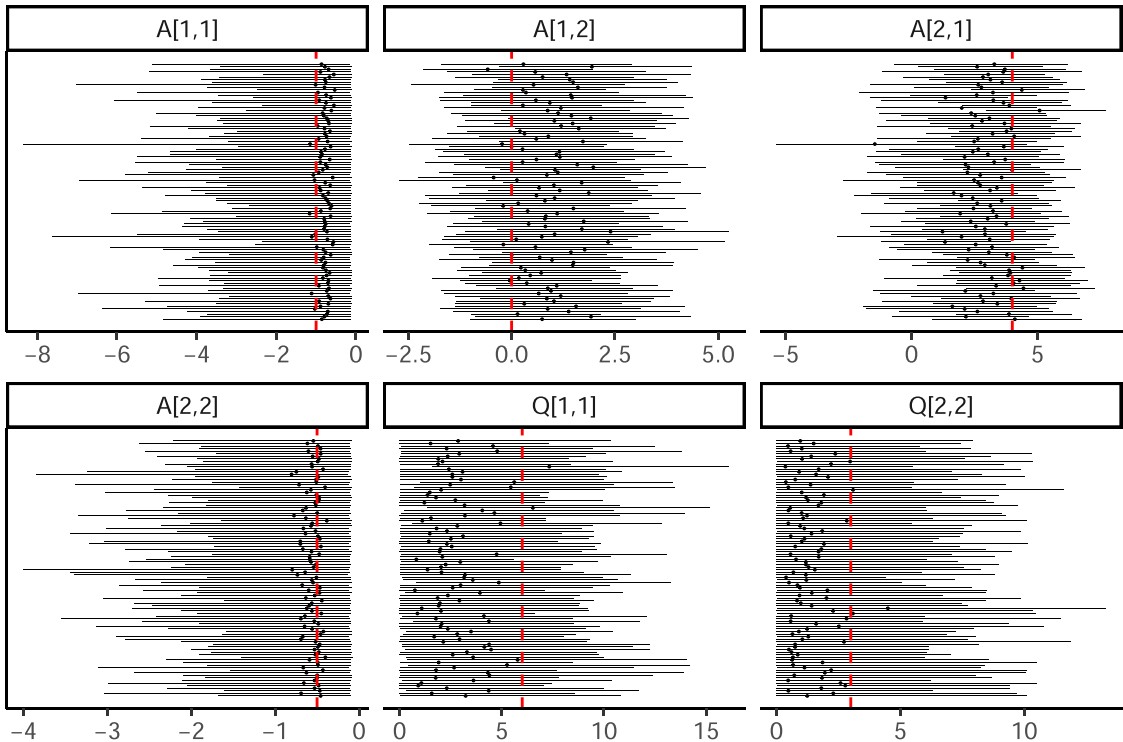

**Extended Data Fig. 4 | Results of simulations of the Bayesian coevolutionary model.** Each row represents model results fitted to 100 different simulated datasets. Fixed 'true' parameter values for these 100 simulated datasets are displayed as red dashed lines. The A matrix captures the effects of selection, and the diagonal of the Q matrix captures the effects of drift. For each model result, points are median posterior parameter values, and lines are 95% credible intervals. The plots show that the red dashed lines reliably fall within the 95% credible intervals, suggesting that the model adequately captures true parameter values.

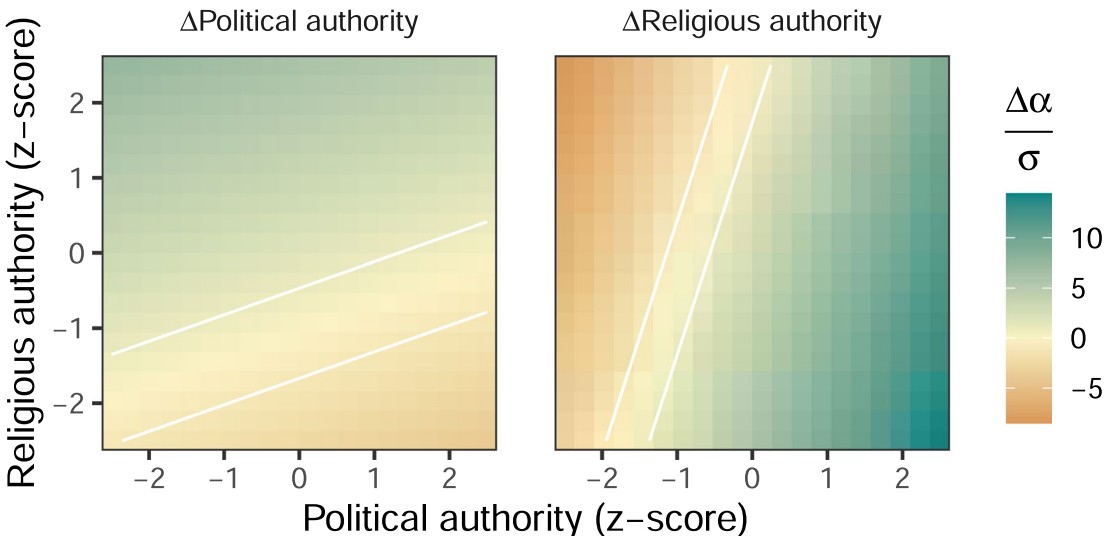

**Extended Data Fig. 5 | Selection gradients for political authority (left) and religious authority (right) from the coevolutionary model, given different combinations of trait levels.** Selection (Δα) is scaled by the strength of drift (σ). Green indicates positive selection (that is for higher rates of authority) and orange indicates negative selection (that is for lower rates of authority). For example, the bottom right corner of the right plot shows that when political authority is high and religious authority is low, there is positive selection on religious authority. White lines encompass areas of the trait space where absolute values are less than 1, indicating that change due to stochastic drift is greater than change due to selection. Trait levels were standardised by the median and median absolute deviation.

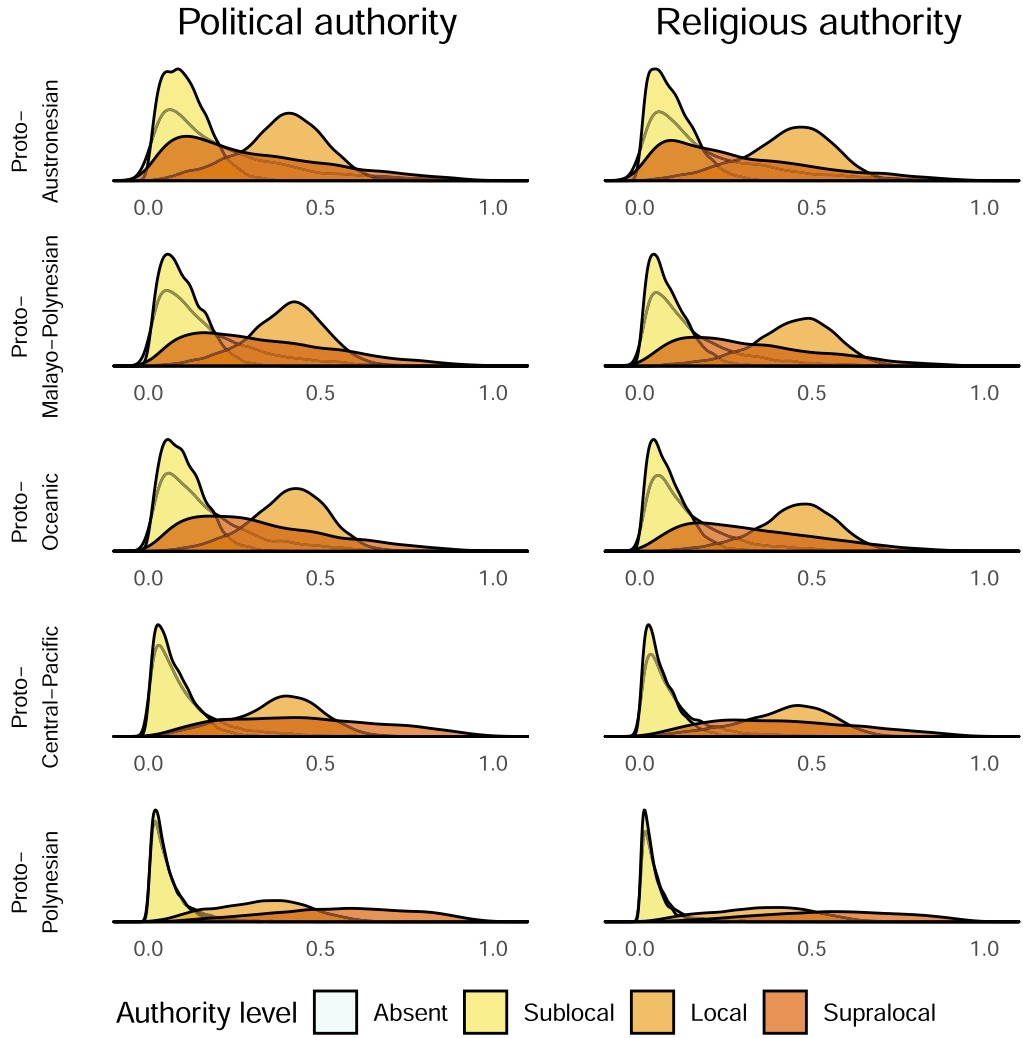

**Extended Data Fig. 6 | Posterior density plots showing the relative probabilities of each authority level at ancestral nodes on the Austronesian language phylogeny, for both political (left) and religious (right) authority.** The x-axis indicates the probability of each authority level, and the y-axis indicates the posterior density. At the root of the phylogeny (Proto-Austronesian), the highest posterior weighting is on local political and religious authority. However, for a more recent node like Proto-Polynesian, the model predicts that supralocal political and religious authority was more likely. For details of the placement of ancestral nodes, see full phylogeny in Fig. 3.

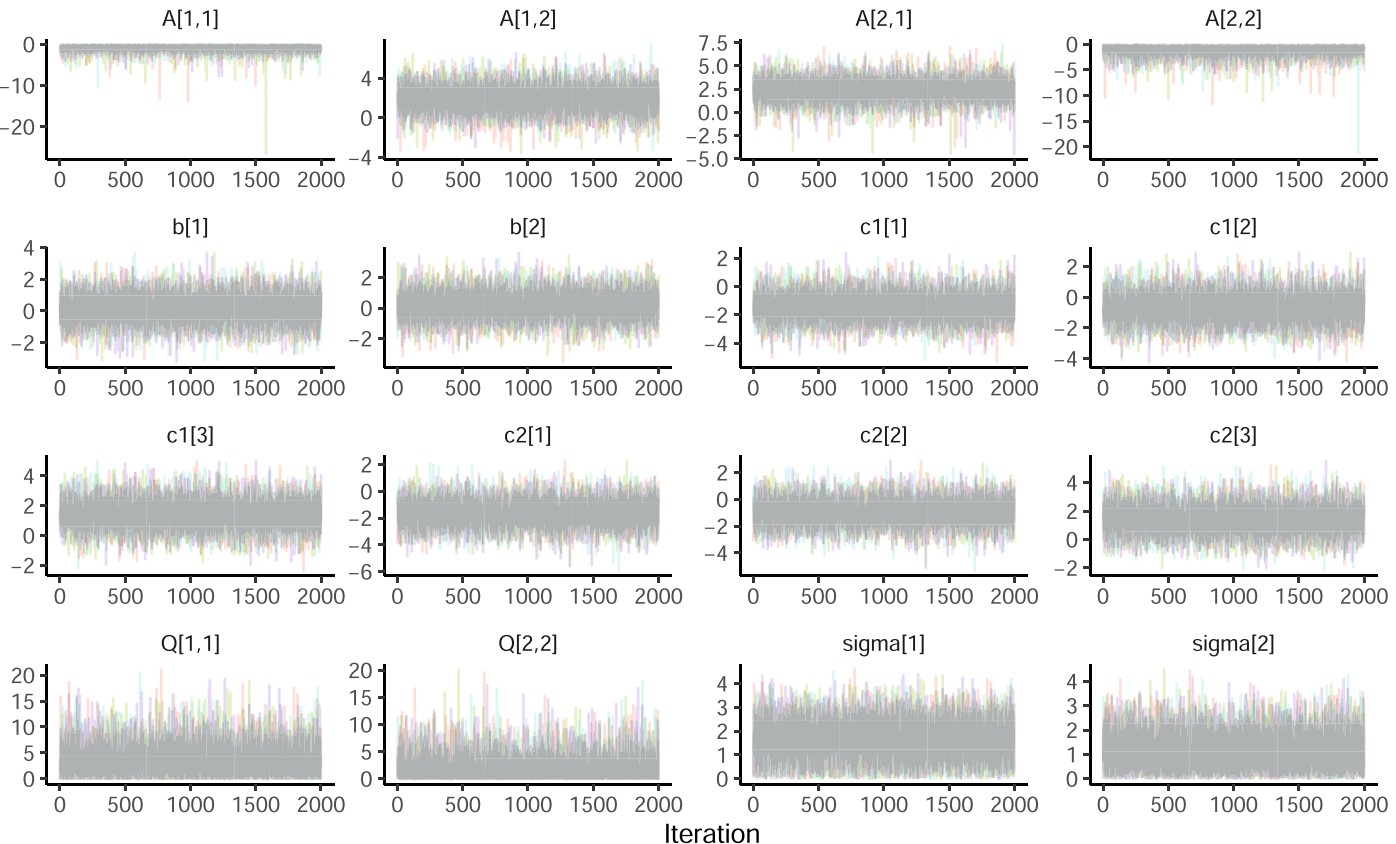

**Extended Data Fig. 7 | Trace plots for the main parameters in the Bayesian coevolutionary model.** For all of these parameters, $\hat{R} \leq 1.01$ and the number of effective samples was greater than 100 times the number the number of chains, suggesting that the model converged normally.

| | |
|---|---|

# Reporting Summary

## Statistics

For all statistical analyses, confirm that the following items are present in the figure legend, table legend, main text, or Methods section.

| n/a | Confirmed | |
|---|---|---|
| ☐ | ☒ | The exact sample size ($n$) for each experimental group/condition, given as a discrete number and unit of measurement |
| ☒ | ☐ | A statement on whether measurements were taken from distinct samples or whether the same sample was measured repeatedly |
| ☒ | ☐ | The statistical test(s) used AND whether they are one- or two-sided<br>*Only common tests should be described solely by name; describe more complex techniques in the Methods section.* |
| ☒ | ☐ | A description of all covariates tested |
| ☒ | ☐ | A description of any assumptions or corrections, such as tests of normality and adjustment for multiple comparisons |
| ☐ | ☒ | A full description of the statistical parameters including central tendency (e.g. means) or other basic estimates (e.g. regression coefficient) AND variation (e.g. standard deviation) or associated estimates of uncertainty (e.g. confidence intervals) |
| ☒ | ☐ | For null hypothesis testing, the test statistic (e.g. $F$, $t$, $r$) with confidence intervals, effect sizes, degrees of freedom and $P$ value noted<br>*Give P values as exact values whenever suitable.* |
| ☐ | ☒ | For Bayesian analysis, information on the choice of priors and Markov chain Monte Carlo settings |
| ☒ | ☐ | For hierarchical and complex designs, identification of the appropriate level for tests and full reporting of outcomes |
| ☒ | ☐ | Estimates of effect sizes (e.g. Cohen's $d$, Pearson's $r$), indicating how they were calculated |

*Our web collection on statistics for biologists contains articles on many of the points above.*

## Software and code

Policy information about availability of computer code

| Data collection | Not applicable - no software used |
|---|---|
| Data analysis | BayesTraitsV3.0 (Multistate), R v4.0.2, (packages: ape, bayestestR, brms, geiger, rstan). Code and command files for all phylogenetic analyses are provided on the OSF (https://osf.io/cm53v/). |

For manuscripts utilizing custom algorithms or software that are central to the research but not yet described in published literature, software must be made available to editors and reviewers. We strongly encourage code deposition in a community repository (e.g. GitHub). See the Nature Portfolio guidelines for submitting code & software for further information.

## Data

Policy information about availability of data

All manuscripts must include a data availability statement. This statement should provide the following information, where applicable:

- Accession codes, unique identifiers, or web links for publicly available datasets
- A description of any restrictions on data availability
- For clinical datasets or third party data, please ensure that the statement adheres to our policy

Data are publicly available on Pulotu (pulotu.com), as well as the OSF (https://osf.io/cm53v/).

# Field-specific reporting

Please select the one below that is the best fit for your research. If you are not sure, read the appropriate sections before making your selection.

☐ Life sciences  ☒ Behavioural & social sciences  ☐ Ecological, evolutionary & environmental sciences

For a reference copy of the document with all sections, see nature.com/documents/nr-reporting-summary-flat.pdf

# Behavioural & social sciences study design

All studies must disclose on these points even when the disclosure is negative.

| | |
|---|---|
| Study description | This is a cultural phylogenetic study that uses quantitative ethnographic data coded from qualitative sources such as ethnographies, and incorporates previous work on the phylogenetic relationships between the societies in the sample. The evolution of cultural traits is modelled under different assumptions. |
| Research sample | The societies in the sample are 97 Austronesian-speaking peoples represented in the ethnographic literature. This sample was chosen because Austronesian language relationships are well-understood, and Austronesian societies are diverse and well-documented. |
| Sampling strategy | Societies chosen based on the availability of ethnographic sources and information on their phylogenetic relationships. We took a pre-existing reconstruction of the Austronesian language family as our starting point. This reconstruction (by Gray, Drummond and Greenhill, 2009) included 400 languages from all major subgroups of the Austronesian language family. We included every language that could be linked to a society for which reliable ethnographic data existed. No power analyses were conducted, but given this sampling strategy it is unlikely that the sample could have been much enlarged. |
| Data collection | The dataset was created by consulting ethnographic materials and coding the variables of interest (initially in a word document, later in an Excel spreadsheet) using pre-specified criteria. The researcher was not blind to the study hypotheses. |
| Timing | A preliminary version of the dataset was coded in June 2017, but revisions continued to be made until May 2021 as the variables of interest evolved and more ethnographic materials were read. |
| Data exclusions | Languages that could not be matched to a society for which adequate ethnographic data were available were excluded. Languages that were spoken by one of the societies in the sample but did not have the largest number of speakers according to Ethnologue were also excluded, based on the requirement of only one language per society. Our final sample included 97 languages out of the 400 in Gray, Drummond and Greenhill (2009), meaning that 303 were 'excluded'. |
| Non-participation | No participants were involved in the study. |
| Randomization | We aimed to include all societies that could be matched to a language in the tree and for which detailed ethnographic data was available, making randomisation inapplicable. |

# Reporting for specific materials, systems and methods

We require information from authors about some types of materials, experimental systems and methods used in many studies. Here, indicate whether each material, system or method listed is relevant to your study. If you are not sure if a list item applies to your research, read the appropriate section before selecting a response.

## Materials & experimental systems

| n/a | Involved in the study |
|---|---|
| ☒ | ☐ Antibodies |
| ☒ | ☐ Eukaryotic cell lines |
| ☒ | ☐ Palaeontology and archaeology |
| ☒ | ☐ Animals and other organisms |
| ☒ | ☐ Human research participants |
| ☒ | ☐ Clinical data |
| ☒ | ☐ Dual use research of concern |

## Methods

| n/a | Involved in the study |
|---|---|
| ☒ | ☐ ChIP-seq |
| ☒ | ☐ Flow cytometry |
| ☒ | ☐ MRI-based neuroimaging |

