## [Peer Review File · Nature Human Behaviour]

Peer Review Information

Journal: Nature Human Behaviour

Manuscript Title: Coevolution of Religious and Political Authority in Austronesian Societies

Corresponding author name(s): Oliver Sheehan

Reviewer Comments & Decisions:

Decision Letter, initial version:

18th November 2020

Dear Dr Sheehan,

Thank you once again for your manuscript, entitled "The Coevolution of Religious and Political Authority in Austronesian Societies", and for your patience during the peer review process.

Your Article has now been evaluated by 2 referees. A third reviewer was unable to send in their report; if they do send us further comments I will of course pass them on to you. You will see from the reviewer comments copied below that, although they find your work of potential interest, they have raised quite substantial concerns. In light of these comments, we cannot accept the manuscript for publication, but would be interested in considering a revised version if you are willing and able to fully address reviewer and editorial concerns.

We hope you will find the referees' comments useful as you decide how to proceed. If you wish to submit a substantially revised manuscript, please bear in mind that we will be reluctant to approach the referees again in the absence of major revisions. We are committed to providing a fair and constructive peer-review process. Do not hesitate to contact us if there are specific requests from the reviewers that you believe are technically impossible or unlikely to yield a meaningful outcome.

To guide the scope of the revisions, the editors discuss the referee reports in detail within the team, including with the chief editor, with a view to (1) identifying key priorities that should be addressed in revision and (2) overruling referee requests that are deemed beyond the scope of the current study. We hope that you will find the prioritised set of referee points to be useful when revising your study. Please do not hesitate to get in touch if you would like to discuss these issues further.

1) Reviewer 1 raises several fundamental concerns relating to your data and analytical approach, and the extent to which these can support your claims. We ask that you address each of these in full, in particular please:

a) Run your analysis scripts on simulated datasets using different scenarios as recommended by the

Reviewer to substantiate your conclusions;

b) Fully address the effect of hidden variables in the analysis itself, as recommended by the Reviewer to provide more robust evidence of causality.

c) Provide transparent and replicable details of both your coding and statistical methodology.

2) Reviewer 3 makes several suggestions for improving the text of the manuscript; please also address these in full.

Finally, your revised manuscript must comply fully with our editorial policies and formatting requirements. Failure to do so will result in your manuscript being returned to you, which will delay its consideration. To assist you in this process, I have attached a checklist that lists all of our requirements. I have also attached a template manuscript file that exemplifies our policies and formatting requirements. If you have any questions about any of our policies or formatting, please don't hesitate to contact me.

If you wish to submit a suitably revised manuscript we would hope to receive it within 6 months. We understand that the COVID-19 pandemic is causing significant disruptions which may prevent you from carrying out the additional work required for resubmission of your manuscript within this timeframe. If you are unable to submit your revised manuscript within 6 months, please let us know. We will be happy to extend the submission date to enable you to complete your work on the revision.

- Include a "Response to the editors and reviewers" document detailing, point-by-point, how you addressed each editor and referee comment. If no action was taken to address a point, you must provide a compelling argument. This response will be used by the editors to evaluate your revision and sent back to the reviewers along with the revised manuscript.
- Highlight all changes made to your manuscript or provide us with a version that tracks changes.

[REDACTED]

Thank you for the opportunity to review your work. Please do not hesitate to contact me if you have any questions or would like to discuss the required revisions further.

Sincerely,

Charlotte Payne
Editor
Nature Human Behaviour

Reviewer expertise:

Reviewer #1: Evolution of religion; Evolution of complex societies

Reviewer #3: Evolution of complex societies; Bayesian phylogenetic methods

REVIEWER COMMENTS:

Reviewer #1:

Remarks to the Author:

This is an ambitious article addressing a very interesting and important question in cultural evolution: the causal relationship between religious and political evolution. Existing theories propose a variety of alternatives: causal precedence of one or another form of authority, co-evolution (mutual causality), or no causal relationship (with correlation arising as a result of sharing causal factors). Untangling such issues of causality, by rejecting some theories in favor of others, is an exciting new direction in cultural evolution and history. It raises also important questions about how we resolve causality in cultural evolution.

An additional strength of the article is its deployment of the Austronesian dataset, which has several unique features facilitating these types of investigations. Most importantly, the process of cultural and linguistic evolution in Austronesia is well represented by the tree model. Cultural phylogenetic methodology is, thus, uniquely suited to this group. Using this methodology to resolve questions of causality is a brilliant idea.

At the same time, while I support publication, the article, in my opinion, require a major revision, primarily focusing on the data and the statistical methods. Generally speaking, the description of methodology is rather opaque, and much more detail needs to be supplied to buttress the conclusions. Here are the specific questions.

1. The connection between ethnographic descriptions and coded data needs to be made much more explicit. Supplementary Table 1 simply lists the “naked” codes and only provides the general references. This is not enough; as anybody, who has converted narrative descriptions into data, knows, this is not a straightforward process. Furthermore, I am a bit surprised that all codes are neatly resolved—there are no missing data, no uncertainty, no disagreement between sources. This is an unusual situation. What the authors need to do is buttress all codes with some kind of explanation for how the code was extracted from the ethnography. This is usually done by quoting a paragraph from the source, or summarizing a lengthy description in the source with a short paragraph, and referencing, of course, to the page in the source. I am sure the authors already have a set of notes along these lines. These notes should be cleaned up and published so that others could examine the coding decisions the authors made, and perhaps suggest alternatives.

2. The statistical methodology is similarly opaque. There is very little detail, beyond citing Mead and Pagel's Bayes Traits manual. I tried to follow the link to the analysis scripts that the authors provide, but it is apparently password-protected. Overall, the statistical model has the feeling of a black box. As I said above, the general idea is brilliant, but does it work specifically for the kinds of questions that the authors want to answer? The data set is not huge—97 binary values for the response and 97 for the predictor variable. That's only 194 bits of information. Is this enough to get strong answers? I don't know. The usual approach in this situation is to simulate artificial data, introducing a variety of complicating factors, then run the analysis scripts on it, to see if they work. I strongly urge the authors to do such tests with simulated data and plausible models of causation.

3. A related question is why only use binary data in each analysis? It seems more natural to me to use the quantitative variables (ranging from 0 to 3) in the analysis, rather than binary ones, which should increase the information content of the dataset, and result in better statistical power.

4. Perhaps the most serious statistical issue, with which the authors don't engage, or even acknowledge, is the hidden variable problem aka omitted variable bias. Statisticians and philosophers of science have long known that inferring causality from correlational data is a thorny problem with multiple pitfalls. One particular problem is that a high correlation between X and Y doesn't necessarily imply a causal relationship between these two variables. It is possible that the causal relationship is $Z \rightarrow X$ and $Z \rightarrow Y$, with Z being the hidden variable. Because both X and Y are cased by Z, in the data they will exhibit strong correlations. Here's a good article that describes this problem, and shows that the hidden variable problem may actually reverse the sign of the relationship:

Eff EA and Routon PW. (2012) Farming and Fighting: An Empirical Analysis of the Ecological-Evolutionary Theory of the Incidence of Warfare. *Structure and Dynamics* 5: 1-33.

Eff and Routon also explain why collapsing a variable with multiple discrete values into a binary one is not a good idea (it reduces variance, that is, reduces the information content of a dataset). Actually, the authors themselves bring up the specter of an omitted variable, when they talk about the possible effect of group size on political and religious authority in the Discussion. The appropriate way of dealing with this problem is to bring any such possible hidden variables explicitly into the analysis.

Please retain my signature
Peter Turchin

Reviewer #2:
None

Reviewer #3:
Remarks to the Author:

This is a remarkably interesting article, thematically in dialogue with classic anthropological debates about complexity, but also with current discussions about kingship and state politics. Although I am very enthusiastic about the fresh air brought by the approach of cultural phylogenetics to classic anthropological issues through the last 10-20 years, it is starting to grow old as long as the very same

approach (not only theory and methods, but even datasets!) are been used again and again without further innovation other than apply them to test a new hypothetical “evolutionary sequence”. However, this manuscript has potential to present a more original take if the authors rewrite the theoretical foundations of the introduction: Theory lacks a more updated approach (maybe applying all what we have learn so far from cultural phylogenetics!) not relying in some ill-defined and questionable idea of directional complexity. The very same results of this study, and most studies using these approaches are probably among the best evidence in anthropology against such notions. Also, the introduction presents a somewhat Eurocentric view of the cut between political and religious power, even though in fact the reported results are largely discrediting that notion. Considering all that, I would rather reduce or exclude most of the first part of the article: It is speculative, not informative, and heavily biased. If time or allotted space do not allow for a renewed theoretical standpoint, maybe can be replaced with a much shorter summary of the problem and lit review.

My other comments are very minor and referred to specific parts of the text:

- Ln 57-67. The paragraph confuses duration with antiquity: A social institution may appear early in the archaeological record but have brief or unstable duration and hence not been old. I presume that is not what the authors meant.
- Ln 89. I suggest adding more details about how the extent of differentiation was coded and defined. It would be even better if the employed criteria is mentioned in the main text (at least briefly) rather than entirely as supplementary material.
- The results presented from Ln 136 onwards would be much nicely explained with a summary table showing most relevant results and statistics for each model (ideally in the main text, if that is acceptable by editorial guidelines)
- The discussion in the paragraph starting at Ln 263: Non sequitur “The fact that all or almost all early states had combined structures of political authority suggest that if there is a sustained trend towards differentiation, this trend emerges only in societies that already reached the state level”. The statement will be benefited from further explanation, which should clarify what they mean by “already reached state level”, particularly after reporting a whole study that seems to disproof the notion that stages are sequentially reached.
- Ln 273, the sentence “even among industrial states” supposes that, for some unmentioned reason, industrialisation should reduce the heterogeneity of the relationship between religion and politics. The claim is presented without any evidence or rationale. If the authors really mean that, they should present such evidence and rationale, and at least discuss Max Weber’s idea of protestant ethics.
- Ln 294. A sentence explaining why and how the purging was done is needed.

In summary, this is a very interesting article and the scientific community would be benefited from its publication. I cannot point out substantial flaws. However, I think it would be a much better article if some of these comments are considered, particularly those about the introduction.

Dear Editor and Reviewers,

We thank you for your detailed and insightful comments. Below is a point-by-point response to reviewers, including an explanation of how we have revised our manuscript to incorporate this feedback.

Reviewer 1

This is an ambitious article addressing a very interesting and important question in cultural evolution: the causal relationship between religious and political evolution. Existing theories propose a variety of alternatives: causal precedence of one or another form of authority, co-evolution (mutual causality), or no causal relationship (with correlation arising as a result of sharing causal factors). Untangling such issues of causality, by rejecting some theories in favor of others, is an exciting new direction in cultural evolution and history. It raises also important questions about how we resolve causality in cultural evolution.

An additional strength of the article is its deployment of the Austronesian dataset, which has several unique features facilitating these types of investigations. Most importantly, the process of cultural and linguistic evolution in Austronesia is well represented by the tree model. Cultural phylogenetic methodology is, thus, uniquely suited to this group. Using this methodology to resolve questions of causality is a brilliant idea.

We thank the reviewer for these positive comments about the paper.

At the same time, while I support publication, the article, in my opinion, require a major revision, primarily focusing on the data and the statistical methods. Generally speaking, the

description of methodology is rather opaque, and much more detail needs to be supplied to buttress the conclusions

As requested, and as outlined below, we have indeed conducted a major revision of the paper, clarifying the data and statistical methods, and deploying a novel analysis technique that allows us to examine the co-evolution of religious and political authority without reducing the data to binary traits.

The connection between ethnographic descriptions and coded data needs to be made much more explicit. Supplementary Table 1 simply lists the ‘naked’ codes and only provides the general references. This is not enough; as anybody, who has converted narrative descriptions into data, knows, this is not a straightforward process. Furthermore, I am a bit surprised that all codes are neatly resolved—there are no missing data, no uncertainty, no disagreement between sources. This is an unusual situation. What the authors need to do is buttress all codes with some kind of explanation for how the code was extracted from the ethnography. This is usually done by quoting a paragraph from the source, or summarizing a lengthy description in the source with a short paragraph, and referencing, of course, to the page in the source. I am sure the authors already have a set of notes along these lines. These notes should be cleaned up and published so that others could examine the coding decisions the authors made, and perhaps suggest alternatives.

We have addressed this comment by adding our coding decisions and their justifications to the database ‘Pulotu’¹. Although many of the societies in our sample corresponded to those already on Pulotu, 21 societies needed to be added. This data can be viewed at pulotu.com by navigating to the ‘compare cultures’ tab and selecting the variables of interest (coding justifications for each society can be viewed by selecting the blue ‘comment’ icons on the far right). In the process of importing the new data and adding societies to Pulotu, we made some changes to the original

dataset. Some coding decisions were altered, and a handful of societies were removed from our original sample and replaced with better-documented ones. We note that despite these changes to the dataset and other methodological changes (described below), our results are largely unchanged.

The statistical methodology is similarly opaque. There is very little detail, beyond citing Mead and Pagel's Bayes Traits manual. I tried to follow the link to the analysis scripts that the authors provide, but it is apparently password-protected.

We apologise for neglecting to make the Open Science Framework page for this paper public. This was a mistake and has now been remedied.

Overall, the statistical model has the feeling of a black box. As I said above, the general idea is brilliant, but does it work specifically for the kinds of questions that the authors want to answer? The data set is not huge—97 binary values for the response and 97 for the predictor variable. That's only 194 bits of information. Is this enough to get strong answers? I don't know. The usual approach in this situation is to simulate artificial data, introducing a variety of complicating factors, then run the analysis scripts on it, to see if they work. I strongly urge the authors to do such tests with simulated data and plausible models of causation ... A related question is why only use binary data in each analysis? It seems more natural to me to use the quantitative variables (ranging from 0 to 3) in the analysis, rather than binary ones, which should increase the information content of the dataset, and result in better statistical power.

We have invested a considerable amount of time and energy in addressing these particular comments, and ultimately brought in two additional coauthors, Erik Ringen and Scott Claessens, to assist with implementing a new analytical approach. This method, outlined by Ringen and colleagues² allowed us to model the coevolution of ordinal variables, which had previously not

been possible. We ran simulations using this new method, which are reported in our latest draft. The model performed well on simulated data.

Perhaps the most serious statistical issue, with which the authors don't engage, or even acknowledge, is the hidden variable problem aka omitted variable bias. Statisticians and philosophers of science have long known that inferring causality from correlational data is a thorny problem with multiple pitfalls. One particular problem is that a high correlation between X and Y doesn't necessarily imply a causal relationship between these two variables. It is possible that the causal relationship is $Z \rightarrow X$ and $Z \rightarrow Y$, with Z being the hidden variable. Because both X and Y are caused by Z , in the data they will exhibit strong correlations. Here's a good article that describes this problem, and shows that the hidden variable problem may actually reverse the sign of the relationship:

Eff EA and Rouston PW. (2012) Farming and Fighting: An Empirical Analysis of the Ecological-Evolutionary Theory of the Incidence of Warfare. Structure and Dynamics 5: 1-33.

Eff and Rouston also explain why collapsing a variable with multiple discrete values into a binary one is not a good idea (it reduces variance, that is, reduces the information content of a dataset).

Actually, the authors themselves bring up the specter of an omitted variable, when they talk about the possible effect of group size on political and religious authority in the Discussion. The appropriate way of dealing with this problem is to bring any such possible hidden variables explicitly into the analysis.

We agree that a 'third variable' is consistent with the reciprocal pattern of coevolution that we observed, and have added a new paragraph to the discussion making this point more explicit. While incorporating a third variable would be technically possible using the new method that we

have adopted, it is by no means obvious which variable to choose, and the most plausible candidates present difficulties. Perhaps the most obvious third variable is ‘authority’ in a more general sense, but investigating the coevolution of three variables, one of which is a broader category encompassing the other two, would be conceptually and methodologically thorny. The same problem would apply to the even broader phenomenon of social complexity³, which is another plausible candidate. What we have done to address this point is to incorporate geographic distance into the model. Geographic distance can serve as a proxy for a range of unmodelled variables, particularly environmental ones⁴. We note that the potential effects of any hidden variable do not mean we cannot say anything about the causal processes involved. The reciprocal co-evolution model we find support for is not compatible with a one-way relationship between the variables - either that religious authority is predicted by political authority but political authority is not predicted by religious authority, or that political authority is predicted by religious authority but religious authority is not predicted by political authority – and is also incompatible with political and religious authority being unconnected causally.

Reviewer 3

This is a remarkably interesting article, thematically in dialogue with classic anthropological debates about complexity, but also with current discussions about kingship and state politics.

We thank the reviewer for these positive comments.

Although I am very enthusiastic about the fresh air brought by the approach of cultural phylogenetics to classic anthropological issues through the last 10-20 years, it is starting to grow old as long as the very same approach (not only theory and methods, but even datasets!) are been used again and again without further innovation other than apply them to test a new hypothetical “evolutionary sequence”. However, this manuscript has potential to present a more

original take if the authors rewrite the theoretical foundations of the introduction: Theory lacks a more updated approach (maybe applying all what we have learn so far from cultural phylogenetics!) not relying in some ill-defined and questionable idea of directional complexity. The very same results of this study, and most studies using these approaches are probably among the best evidence in anthropology against such notions.

We assume that ‘directional complexity’ refers in this context either to (a) a tendency for societies to become more complex over time, or (b) the idea that there are directional relationships between the components or drivers of social complexity. We do not believe that either (a) or (b) correspond to outdated or discredited views. Social complexity may or may not be a unitary construct, and the details of how it evolves are still not fully resolved. However, the evidence that human societies have tended to become more complex over the last 10,000 years is clear (see e. g. ⁵), and includes the results of cultural phylogenetic studies (see e. g. ⁶). Turning to (b), there is also no a priori reason why the cultural traits associated with complexity cannot have directional relationships, or why cultural phylogenetic methods cannot uncover these relationships. Cultural phylogenetic studies often find evidence of reciprocal causation, but not always. Currie and Mace⁶ found a directional relationship between class stratification and political complexity: the presence of class stratification increased the likelihood that a society would gain political levels beyond the local community, but having political levels beyond the local community did not increase the likelihood of gaining class stratification. More recently Ringen and colleagues², whose method we use in our revised version of the study, found a directional relationship between resource intensification (RI) and technological and social differentiation (TSI): increases in RI led to increases in TSI, but increases in TSI did not lead to increases in TSI.

Also, the introduction presents a somewhat Eurocentric view of the cut between political and religious power, even though in fact the reported results are largely discrediting that notion.

Considering all that, I would rather reduce or exclude most of the first part of the article: It is speculative, not informative, and heavily biased. If time or allotted space do not allow for a renewed theoretical standpoint, maybe can be replaced with a much shorter summary of the problem and lit review.

We disagree strongly with the characterisation of the first part of our article as ‘biased’. It is a summary of the views on political and religious authority that can be found in the academic literature, not of our own views. We explicitly acknowledge that one of these views is that religious and political authority cannot be meaningfully separated⁷. We are reluctant to ‘reduce or exclude’ this section as it provides crucial context and justification for our study.

Ln 57-67. The paragraph confuses duration with antiquity: A social institution may appear early in the archaeological record but have brief or unstable duration and hence not been old. I presume that is not what the authors meant.

This is a fair point, but we would argue that if one trait appears earlier than another in the archaeological record, in the absence of other evidence it is reasonable to characterise the trait that appears earlier as older. In any case, we go on to note that the archaeological record is too fragmentary to resolve the questions at hand.

Ln 89. I suggest adding more details about how the extent of differentiation was coded and defined. It would be even better if the employed criteria is mentioned in the main text (at least briefly) rather than entirely as supplementary material.

We provide a detailed explanation of how the coding of differentiation (as well as the other variables) was achieved in the Results and Methods sections, not the Supplementary Material.

The results presented from Ln 136 onwards would be much nicely explained with a summary table showing most relevant results and statistics for each model (ideally in the main text, if that is acceptable by editorial guidelines).

We appreciate this advice, but we have implemented a new method and no longer include the constrained analyses described in the paragraph mentioned above.

The discussion in the paragraph starting at Ln 263: Non sequitur “The fact that all or almost all early states had combined structures of political authority suggest that if there is a sustained trend towards differentiation, this trend emerges only in societies that already reached the state level”. The statement will be benefited from further explanation, which should clarify what they mean by “already reached state level”, particularly after reporting a whole study that seems to disprove the notion that stages are sequentially reached.

By ‘reach[ing] the state level’ we are simply referring to the emergence of states in societies that previously lacked them. That the earliest state-level societies evolved from non-state societies is not controversial. Our results did not support long-term trends towards differentiation or fusion of religious and political authority, but this does not mean that our study disproves the notion that stages are reached sequentially.

Ln 273, the sentence “even among industrial states” supposes that, for some unmentioned reason, industrialisation should reduce the heterogeneity of the relationship between religion and politics. The claim is presented without any evidence or rationale. If the authors really mean that, they should present such evidence and rationale, and at least discuss Max Weber’s idea of protestant ethics.

We agree and have removed the relevant sentence.

Ln 294. A sentence explaining why and how the purging was done is needed.

We have added the text ‘Because our method required only one language per society’ to the previous sentence, which addresses the ‘why’ part of the comment above. We believe that we have already adequately explained ‘how’ the pruning was done by providing the criteria by which one language was chosen for each society, and the names of the R packages that we used to remove the other languages from the sample of phylogenies.

We thank you all again for your time and for your thoughtful and helpful comments, and look forward to hearing your decision.

Best wishes,

Oliver Sheehan, Joseph Watts, Russell Gray, Joseph Bulbulia, Scott Claessens, Erik Ringen, and Quentin Atkinson

References

1. Watts, J. et al. Pulotu: Database of Austronesian supernatural beliefs and practices. *PloS One* **10**, e0136783 (2015).
2. Ringen, E. J., Martin, J.S., & Jaeggi, A. V. Novel phylogenetic methods reveal that resource-use intensification drives the evolution of ‘complex’ societies. *EcoEvoRxiv*, <https://ecoevorxiv.org/wfp95/> (2021).

3. Turchin, P., et al. Quantitative historical analysis uncovers a single dimension of complexity that structures global variation in human social organization. *Proceedings of the National Academy of Sciences* **115**, E144-E151 (2018).
4. Freckleton, R. P., & Jetz, W. Space versus phylogeny: Disentangling phylogenetic and spatial signals in comparative data. *Proceedings of the Royal Society B: Biological Sciences* **276**, 21-30 (2009).
5. Marcus, J. The archaeological evidence for social evolution. *Annual Review of Anthropology*, **37**, 251-266 (2008).
6. Currie, T. E., & Mace, R. Mode and tempo in the evolution of socio-political organization: reconciling ‘Darwinian’ and ‘Spencerian’ evolutionary approaches in anthropology. *Philosophical Transactions of the Royal Society B: Biological Sciences*, **366**, 1108-1117 (2011).
7. Bloch, M. Why religion is nothing special but is central. *Philos. Trans. R. Soc. Lond. B: Biol. Sci.* **363**, 2055-2061 (2008).

Decision Letter, first revision:

7th April 2022

Dear Dr. Sheehan,

Thank you for submitting your revised manuscript "Coevolution of Religious and Political Authority in Austronesian Societies" (NATHUMBEHAV-200812197A). It has now been seen by the original referees. Reviewer 1 mentioned in confidential comments to editors that they are in favour of acceptance, and Reviewer 3's comments are below. As you can see, the reviewers find that the paper has improved in

revision. We will therefore be happy in principle to publish it in Nature Human Behaviour, pending minor revisions to satisfy the referees' final requests and to comply with our editorial and formatting guidelines.

Please note that we will ask you to address Reviewer 3's concerns in full, including a more in depth discussion of the issues they raise regarding complexity, and clarifying your rationale for requiring only one language per society in your Methods.

We are now performing detailed checks on your paper and will send you a checklist detailing our editorial and formatting requirements within a week. Please do not upload the final materials and make any revisions until you receive this additional information from us.

Sincerely,

Charlotte Payne

Charlotte Payne, PhD
Senior Editor
Nature Human Behaviour

Reviewer #3 (Remarks to the Author):

My recommendations towards trimming and moderate some theoretical claims were not agreed on. I am completely happy with the authors not agreeing with my views and I still consider the article a very good article and worth publishing: indeed, some of these matters may spark fruitful debates around some interesting issues about the idea of social complexity. While I do not want to go point by point arguing with the authors' answers, I would like to paraphrase the main ideas from my previous comments to provide a better explanation of what I meant, in hope to provide the authors a constructive sample of what may also be the view of some eventual readers.

My previous review tried to point out that the introduction had a too frugal discussion of the idea of complexity, a controversial idea in both biology and anthropology, while focusing in labelling some forms of political organisation on a scale where time and complexity run together. I continue to think that neither this manuscript nor its references support that view. Cultural phylogenetics have played a big role discrediting that idea and I also think that this is not what the authors meant. My apologies if the confusion was prompted by my choice of words (directional complexity, Eurocentric, etc.): by all means I agree that there are directional relationships between the components and drivers of social change: Taking into account the terminology used in the articles mentioned in the authors reply I could replace my wording for what Currie and Mace 2011 called "rectilinear" (though historically was called unilineal, a denomination they use for something else).

Most of the articles in the manuscript references and the manuscript itself discredit the 19th century idea that social evolution is necessarily "rectilinear" (to use Currie and Mace terminology). Ln 194. "we found no substantial evidence that either form of authority had causal precedence" Ln 269. "We found

no clear evidence for or against a progression from less differentiated to more differentiated systems of authority", etc.

While the authors "explicitly acknowledge that one of these views is that religious and political authority cannot be meaningfully separated", hard claims such as "political and religious authorities have worked synergistically over millennia to drive the evolution of large-scale societies" or "The scale and complexity of human societies increased immensely during the Holocene. Authority, a form of social power vested in a culturally recognised 'role' or 'office' and exercised over a specific group of people, was one of the key innovations that enabled this transition" suggest, at least to me, otherwise.

Regarding methods, very substantial changes were introduced to the current version of the manuscript resulting in a clearer description of data collation, coding and analyses. Reading the response letter, I felt that my questions and comments about data coding were miscommunicated, but fortunately they were fully addressed in both the article and in the response to the other reviewer, who made the same comment but somehow managed to make him/herself clearer. The changes to methods, particularly changing from binary categorical coding to ordinal coding and the provision of code and data in the OSF page are very welcomed additions.

Other methodological question I had (i.e. how and why pruning was done) was replied quoting from the text that "our method required only one language per society", and mentioned that "names of R packages are provided". I am sorry my comment was not clearly stated: What I meant was why the method requires one language per society? and why the largest number of speakers is deemed as the selection criteria when more than one language is spoken? There are various ways to clarify this depending of what was your rationale: for instance, "using separate codes for societies with more than one language would underestimate the relationship between societies sharing only a fraction of their languages". There are some alternative to what you did (i.e. differential coding, treat each language as a trait present at a value between 0-1 in all societies, and maybe more), there are very good reasons for your choice, but none are mentioned. As with every method, there are also some caveats: choosing a language underestimates the relationship between societies sharing languages other than the most spoken language, and that would be good mentioned.

Author Rebuttal, first revision:

Response to Reviewer #3

We thank Reviewer #3 for the additional comments, which we address below.

My previous review tried to point out that the introduction had a too frugal discussion of the idea of complexity, a controversial idea in both biology and anthropology, while focusing in labelling some forms of political organisation on a scale where time and complexity run together. I continue to think that neither this manuscript nor its references support that view. Cultural

phylogenetics have played a big role discrediting that idea and I also think that this is not what the authors meant. My apologies if the confusion was prompted by my choice of words (directional complexity, Eurocentric, etc.): by all means I agree that there are directional relationships between the components and drivers of social change: Taking into account the terminology used in the articles mentioned in the authors reply I could replace my wording for what Currie and Mace 2011 called “rectilinear” (though historically was called unilineal, a denomination they use for something else).

Most of the articles in the manuscript references and the manuscript itself discredit the 19th century idea that social evolution is necessarily “rectilinear” (to use Currie and Mace terminology). Ln 194. “we found no substantial evidence that either form of authority had causal precedence” Ln 269. “We found no clear evidence for or against a progression from less differentiated to more differentiated systems of authority”, etc.

We thank the reviewer for the clarification as well as the encouraging words. However, we suspect that our interpretations of the literature on social complexity still differ.

The reviewer’s characterisation of the ‘idea of complexity’ as ‘controversial’ is in our view only partly accurate. It is true that there is some disagreement about what exactly social complexity is, and whether or not it is a single construct¹⁻³. However, the idea that socio-political complexity has tended to increase over time - which we assume is what is meant by ‘a scale where time and complexity run together’ - is not at all controversial. The archaeological evidence for this trend is overwhelming⁴ and phylogenetic studies, far from discrediting this view, also support it. Currie and Mace⁵, whom the reviewer cites, found support for both increases and decreases in political complexity having occurred, but also found that increases had been more common. We accept that our paper does not directly support a tendency for social complexity to increase. However, we are interested in how the increase in complexity occurred, not whether it occurred, which we regard as settled.

While the authors “explicitly acknowledge that one of these views is that religious and political authority cannot be meaningfully separated”, hard claims such as “political and religious authorities have worked synergistically over millennia to drive the evolution of large-scale societies” or “The scale and complexity of human societies increased immensely during the Holocene. Authority, a form of social power vested in a culturally recognised ‘role’ or ‘office’ and exercised over a specific group of people, was one of the key innovations that enabled this transition” suggest, at least to me, otherwise.

This point is a fair one. The view that we acknowledged in the paper is more accurately characterized as the view that religion and politics cannot meaningfully be separated in most pre-modern societies, not that they cannot be separated in any circumstances. We have made a slight change to the wording of this section of the paper (lines 66-68) to reflect this point. We stand by our statements that “[t]he scale and complexity of human societies increased immensely during the Holocene’ (see previous point) and that “[a]uthority, a form of social power vested in a culturally recognised ‘role’ or ‘office’ and exercised over a specific group of people, was one of the key innovations that enabled this transition”. The sources that we cite support these claims.

Other methodological question I had (i.e. how and why pruning was done) was replied quoting from the text that “our method required only one language per society”, and mentioned that “names of R packages are provided”. I am sorry my comment was not clearly stated: What I meant was why the method requires one language per society? and why the largest number of speakers is deemed as the selection criteria when more than one language is spoken? There are various ways to clarify this depending of what was your rationale: for instance, “using separate codes for societies with more than one language would underestimate the relationship between societies sharing only a fraction of their languages”. There are some alternative to what you did (i.e. differential coding, treat each language as a trait present at a value between 0-1 in all societies, and maybe more), there are very good reasons for your choice, but none are

mentioned. As with every method, there are also some caveats: choosing a language underestimates the relationship between societies sharing languages other than the most spoken language, and that would be good mentioned.

We thank the reviewer again for the clarification. Our method requires one language per society because it treats languages as proxies for societies. It was not possible to (for example) treat languages as traits of societies for this reason. We could have assigned the same ethnographic datapoint to multiple languages, but choosing one language per society was the more conservative strategy, avoiding pseudo-replication in the data. Our rationale for basing our choice on the number of speakers is that to the extent that language is a proxy for culture, the language that is most widely spoken within a given society will be most representative of the culture of that society. We are unsure what other criterion we could have used.

We understand the reviewer's concern about underestimating the relationship between societies with overlapping language affiliations, but in fact there were no such societies in the sample. Some societies corresponded to multiple languages, but no language corresponded to more than one society. We have reworded the 'phylogenies' section (lines 283-294) to make this point, as well as our reasons for choosing one language per society, clearer.

We thank the reviewer again for these comments and clarifications.

Best wishes,

Oliver Sheehan (and coauthors)

References

1. Chick, G. Cultural complexity: The concept and its measurement. *Cross Cult. Res.*, **31**, 275-307 (1997).
2. Denton, T. Cultural complexity revisited. *Cross Cult. Res.*, **38**, 3-26 (2004).
3. Turchin, P. et al. Quantitative historical analysis uncovers a single dimension of complexity that structures global variation in human social organization. *PNAS*, **115**, E144-E151 (2018).
4. Marcus, J. The archaeological evidence for social evolution. *Annu. Rev. Anthropol.*, **37**, 251-266 (2008).
5. Currie, T. E., & Mace, R. Mode and tempo in the evolution of socio-political organization: Reconciling 'Darwinian' and 'Spencerian' evolutionary approaches in Anthropology. *Philos. Trans. R. Soc. B*, **366**, 1108-1117 (2011).

Final Decision Letter:

Dear Dr Sheehan,

We are pleased to inform you that your Article "Coevolution of Religious and Political Authority in Austronesian Societies", has now been accepted for publication in *Nature Human Behaviour*.

Please note that *Nature Human Behaviour* is a Transformative Journal (TJ). Authors whose manuscript was submitted on or after January 1st, 2021, may publish their research with us through the traditional subscription access route or make their paper immediately open access through payment of an article-processing charge (APC). Authors will not be required to make a final decision about access to their article until it has been accepted. IMPORTANT NOTE: Articles submitted before January 1st, 2021, are not eligible for Open Access publication. Find out more about Transformative Journals

Authors may need to take specific actions to achieve compliance with funder and institutional open access mandates. If your research is supported by a funder that requires immediate open access (e.g. according to Plan S principles) then you should select the gold OA route, and we will direct you to the compliant route where possible. For authors selecting the

subscription publication route, the journal's standard licensing terms will need to be accepted, including self-archiving policies. Those licensing terms will supersede any other terms that the author or any third party may assert apply to any version of the manuscript.

With best regards,

Charlotte Payne

Charlotte Payne, PhD
Senior Editor
Nature Human Behaviour